# Glucocorticoids: Fuelling the Fire of Atherosclerosis or Therapeutic Extinguishers?

**DOI:** 10.3390/ijms22147622

**Published:** 2021-07-16

**Authors:** Clare MacLeod, Patrick W. F. Hadoke, Mark Nixon

**Affiliations:** University/British Heart Foundation Centre for Cardiovascular Science, The Queen’s Medical Research Institute, University of Edinburgh, Edinburgh EH16 4TJ, UK; s1506720@sms.ed.ac.uk (C.M.); patrick.hadoke@ed.ac.uk (P.W.F.H.)

**Keywords:** glucocorticoids, atherosclerosis, inflammation, cardiovascular diseases, glucocorticoid receptor, lipids

## Abstract

Glucocorticoids are steroid hormones with key roles in the regulation of many physiological systems including energy homeostasis and immunity. However, chronic glucocorticoid excess, highlighted in Cushing’s syndrome, is established as being associated with increased cardiovascular disease (CVD) risk. Atherosclerosis is the major cause of CVD, leading to complications including coronary artery disease, myocardial infarction and heart failure. While the associations between glucocorticoid excess and increased prevalence of these complications are well established, the mechanisms underlying the role of glucocorticoids in development of atheroma are unclear. This review aims to better understand the importance of glucocorticoids in atherosclerosis and to dissect their cell-specific effects on key processes (e.g., contractility, remodelling and lesion development). Clinical and pre-clinical studies have shown both athero-protective and pro-atherogenic responses to glucocorticoids, effects dependent upon their multifactorial actions. Evidence indicates regulation of glucocorticoid bioavailability at the vasculature is complex, with local delivery, pre-receptor metabolism, and receptor expression contributing to responses linked to vascular remodelling and inflammation. Further investigations are required to clarify the mechanisms through which endogenous, local glucocorticoid action and systemic glucocorticoid treatment promote/inhibit atherosclerosis. This will provide greater insights into the potential benefit of glucocorticoid targeted approaches in the treatment of cardiovascular disease.

## 1. Introduction

Glucocorticoids are steroid hormones with an essential role in the regulation of numerous physiological processes, including energy utilisation, immune response, blood pressure, mood/memory, and stress responses [1]. Their secretion into the circulation occurs in a pulsatile, circadian manner with peak levels prior to waking, and a nadir during the early stages of sleep. Additionally, there is also evidence of longer infradian rhythms of cortisol excretion suggesting more complex control than just circadian rhythmicity, which may add further complication to accurate measurement of exposure to cortisol [2]. Glucocorticoids are synthesised from cholesterol in the cortical layer of the adrenal gland, specifically the zona fasciculata, and are released in response to stimulation by adrenocorticotropic hormone (ACTH) secreted from the anterior pituitary gland. This release is regulated by feedback control of the hypothalamic pituitary adrenal (HPA) axis. In humans, the primary glucocorticoid is cortisol although they also produce corticosterone, whereas in rodents, which provide primary pre-clinical models for disease, the lack of adrenal CYP17A1 means corticosterone is the sole glucocorticoid [3].

In the circulation, the bioavailability of glucocorticoids is regulated by plasma binding proteins—primarily corticosteroid-binding globulin (CBG), but also albumin [4]. Under normal physiological conditions, 80–90% of glucocorticoids are bound to CBG and 10–15% are bound to albumin, while 2–10% remain unbound or “free”. Only the free fraction is able to diffuse across cell membranes into cells and exert biological activity [5,6]. Glucocorticoids exert the majority of their actions through genomic routes via either the ubiquitously expressed glucocorticoid receptor (GR), or the tissue-specific mineralocorticoid receptor (MR); binding MR with higher affinity [7,8,9].

Glucocorticoids have important roles in physiology and are key in the maintenance of cardiovascular health. This review will focus on the ability of glucocorticoids to regulate atherosclerosis, both in terms of driving the pathogenesis of the disease (pro-atherogenic), and as a potential therapeutic option to limit disease severity (athero-protective). Here we will outline the historical literature and explore the more recent evidence underlying how glucocorticoids influence the development of atheromatous plaques, in particular dissecting the cell-specific mechanisms regulating glucocorticoid action in this disease setting.

## 2. Glucocorticoids and Cardiovascular Risk: More Than an Association?

The ability of glucocorticoids to regulate numerous key physiological processes, in particular the body’s immune response and fuel metabolism, mean they play an important role in the maintenance of cardiovascular health. However, these essential, adaptive responses can become maladaptive upon chronic exposure to elevated levels of glucocorticoids (Table 1). Indeed, there is a well-established correlation between chronic glucocorticoid excess and increased cardiovascular disease (CVD) risk, notably the development of key risk factors including obesity, diabetes, hypertension, and dyslipidaemia [9,10].

The most striking demonstration of the cardiovascular consequences of glucocorticoid excess remains that highlighted in the clinical presentation of Cushing’s syndrome [17]. Individuals with this condition have excess endogenous cortisol production, typically resulting from either pituitary or adrenal tumours, and present multiple symptoms including visceral obesity, insulin resistance and dyslipidaemia—all CVD risk factors [17]. Moreover, analysis has demonstrated that Cushing’s patients have an increased risk of cardiovascular events such as coronary artery disease, myocardial infarction, stroke and heart failure [18,19].

Whilst cases of Cushing’s are extremely rare, affecting between 40 and 70 people out of every million [20], the link between glucocorticoid excess and cardiovascular risk has been identified in other scenarios. Sub-clinical hypercortisolism (also referred to as “subclinical Cushing’s syndrome”, “pre-clinical Cushing’s syndrome”, or “sub-clinical autonomous glucocorticoid hypersecretion”) is an imperfectly defined condition indicated in patients exhibiting mild alteration in cortisol secretion without the symptoms of Cushing’s syndrome [21,22]. It is usually associated with adrenocortical incidetalomas detected during imaging for conditions unrelated to the adrenal gland. Clinical investigations have associated chronic exposure to “sub-clinical” hypercortisolism with a number of complications (including excess weight/obesity, hypertension, and diabetes mellitus) that are risk factors for CVD [22,23]. Indeed, there is some evidence of increased cardiovascular events and mortality in these patients, indicating a possible benefit from medical intervention or adrenalectomy [24]. This is important as adrenal incidentalomas are considerably more common than Cushing’s syndrome (possibly present in 4–7% of adults) [25], and the prevalence of sub-clinical hypercortisolism in these patients has been estimated as 0.2–2.0% [23].

A further link between glucocorticoid excess and the development of cardiovascular disease has been suggested in patients with metabolic syndrome; a condition associated with increased cardiovascular risk [26]. The similarity between the features of metabolic syndrome (a combination of at least three symptoms out of central obesity, hypertension, hyperglycaemia, hypertriglyceridaemia, and low serum high-density lipoprotein) and Cushing’s syndrome, has prompted the suggestion that the former represents a “low cortisol”, or “tissue-specific” form of Cushing’s syndrome [27]. This proposal is based on an improved understanding of the role of glucocorticoid-metabolising enzymes (specifically the 11β-hydroxysteroid dehydrogenases (11β-HSDs)) in regulating activation of corticosteroid receptors in a tissue-specific manner: raising the possibility of tissue-specific glucocorticoid excess through aberrant expression of the 11β-HSD isozymes (e.g., in the liver [28] or central adipose tissue [29]) without elevated circulating levels of the steroid. This has led to the investigation of pharmacological targeting of 11β-HSDs for treatment of metabolic disease.

In addition to the evidence obtained from exposure to elevated endogenous glucocorticoids, the widespread therapeutic use of glucocorticoids to treat inflammatory and autoimmune conditions has also provided supporting evidence for their association with CVD. While effective in alleviating symptoms of asthma, systemic lupus erythematosus (SLE) and rheumatoid arthritis (RA) [30], glucocorticoids do not treat the underlying cause of these diseases, and in many cases patients remain on long-term glucocorticoid treatment to effectively manage their condition. Unless carefully managed, this chronic exposure to supraphysiological doses of glucocorticoids can result in iatrogenic Cushing’s syndrome. Several studies have highlighted that patients administered high dose glucocorticoids have an increased risk of CVD (as reviewed by [31]). A recent analysis of a UK population-based cohort demonstrated an increased risk of cardiovascular events across six different immune pathologies in which patients were being treated with prednisolone [32]. This study demonstrated a dose-dependent effect of prednisolone associated with an increase in the prevalence of cardiovascular events such as acute myocardial infarction and peripheral arterial disease. While this supports previous studies in which similar dose-dependent effects of glucocorticoid treatment were observed to increase the risk of cardiovascular events [33,34], this work highlighted that even long term low dose glucocorticoid treatment, previously considered “safe”, resulted in a doubling of overall CVD risk [32].

Together, the evidence for an association between increased exposure to glucocorticoids and increased CVD risk is clear, yet the question of causality remains. Using Mendelian randomisation and data collected across 17 studies of European ancestry, a recent study tested whether genetically elevated cortisol in the general population is causally associated with a number of cardiovascular risk factors [35]. Importantly, the authors demonstrate that increased cortisol causally increases the risk of ischaemic heart disease and myocardial infarction. However, as alluded to by the authors, their measures of morning plasma cortisol are likely poor surrogates for overall tissue exposure. Indeed, over the years several intricate molecular mechanisms have been identified that regulate the cellular exposure to glucocorticoids. To further assess the impact of glucocorticoids on atherosclerosis per se, a greater knowledge of glucocorticoid action at a cell-specific level is required, in parallel with more robust assessments of glucocorticoid phenotypes.

## 3. Atherosclerosis: A Complex, Multi-Cellular Pathophysiology

Atherosclerosis is a progressive inflammatory disease, characterised by a narrowing of the arteries due to vascular remodelling and formation of atheromatous plaques (Figure 1). The clinical manifestation of atherosclerosis is dependent on the site of plaque formation and the organ affected by impaired blood supply. The development of atherosclerosis in the coronary arteries (coronary heart disease) can cause angina and myocardial infarction, whereas presentation in the brain and cerebral circulation (cerebrovascular disease) can result in a stroke.

In line with our current understanding from pre-clinical models, there are two key events in atherogenesis: endothelial cell damage and vascular remodelling. In particular, the recruitment and activation of circulating immune cells, lipoprotein retention, and the migration and proliferation of vascular smooth muscle cells (vSMCs) are viewed as critical cellular responses. Research has established that initial intimal thickening of arteries is physiological, developing silently throughout an individual’s lifetime, and is evident in young children and even babies less than a year old [36]. However, this adaptive intimal thickening can progress into maladaptive remodelling upon exposure to high circulating levels of cholesterol, specifically low-density lipoproteins (LDL), alongside endothelial cell damage as proposed by the response-to-injury hypothesis [37]. Damage to the endothelium causes increased permeability, allowing LDL to migrate into the sub-endothelial space of the intimal layer and become oxidised (oxLDL) [38]. This accumulation of lipoprotein triggers a cascade of inflammatory cell influx into the intima of the vascular wall. In particular, monocytes are recruited to the sub-endothelial layer by attachment to adhesion molecules including vascular cell adhesion molecule (VCAM-1) and intercellular adhesion molecule (ICAM-1), expressed on the activated endothelial cells following endothelial cell damage [39]. However, there is also evidence of neutrophil infiltration following endothelial cell damage, with studies in pre-clinical models indicating that their levels peak at the early stage of atherosclerosis and fall to almost undetectable levels in developed plaques [40,41]. While other immune cells are also implicated in the development of atherosclerosis, including dendritic cells, mast cells and lymphocytes [42], monocyte recruitment and subsequent differentiation into macrophages within the intima reflects the central role of monocytes/macrophages in plaque formation. Macrophage activation within the vasculature results in the ingestion of local oxLDL, and the subsequent formation of foam cells. The accumulation of these foam cells is a central part of the vascular remodelling in atherosclerosis, and results in a “fatty streak” which can either stabilise, regress or progress into an atherosclerotic plaque.

The proliferation and migration of vSMCs from the media to the intima forms another significant component of vascular remodelling. Under the pro-inflammatory environment, vSMCs undergo a phenotypic switch in which they lose their contractile phenotype, releasing growth factors and matrix metalloproteases (MMPs), and gain a more macrophage and foam cell-like phenotype [38,43]. As these foam cells undergo apoptosis or necrosis, the aggregation of inflammatory cells and cytokines, macrophage-like vSMCs, and dying foam cells, form a necrotic core between the intimal and medial layers causing further plaque instability and inflammation. At this stage, there can be further pathological intimal thickening, and vascular dysfunction due to loss of contractility of the vascular smooth muscle and impaired endothelium-dependent relaxation. A fibrotic cap composed of connective tissue forms, which is postulated to be a protective remodelling mechanism to prevent leakage of the pro-thrombotic material into the circulation [38].

A major aspect of vascular remodelling is the degradation of extracellular matrix (ECM), which occurs as a consequence of protease action. Neutrophil elastase (NE) and MMPs, released from activated neutrophils, macrophages, endothelial cells and vSMCs are key proteases that promote the breakdown of ECM by degrading structural proteins such as collagen and elastin [44,45]. The result of this is structural damage to the vasculature, contributing to plaque rupture and lack of contractility. Of particular interest is a newly identified role for NE in atherosclerosis. The expression of NE is increased both in human plaques and in plaques from murine models of atherosclerosis [46,47], but recent studies have explored a causative role for this protease, demonstrating a pathogenic role in foam cell formation through modulation of cholesterol efflux [46].

In humans, most plaques are stable and remain asymptomatic; however, plaques can rupture resulting in thrombosis and occlusion of arteries. The “stabilisation” of plaques is regarded as any change in the composition in which reduces the plaques vulnerability to rupture or erode. Therefore, identifying characteristics of these “unstable” plaques is crucial. Indeed, stable plaques have thick protective fibrous caps with small necrotic cores with less macrophage infiltration [48]. The question of plaque vulnerability underlines crucial species-specific differences in atherosclerosis and highlights a limitation of pre-clinical research into atherosclerosis. For example, plaques in murine models of atherosclerosis such as ApoE^−/−^ mice can have features of vulnerability, but death caused by plaque rupture is not evident [49].

Additionally, it is important to recognise that approximately 20% of deaths in acute coronary syndrome (ACS) patients can be caused by superficial plaque erosion, which has a distinct mechanism from plaque rupture [50]. Typically, eroded plaques have fewer inflammatory cells, and lack the lipid rich necrotic core associated with vulnerable plaques [50]. The different plaque types produce different thrombi, with eroded plaques producing white thrombus with high platelet content whereas ruptured plaques produce red thrombus with a high lipid and fibrin content [51]. The events that determine whether a plaque erodes, or ruptures are still unclear, although research remains predominantly focused on understanding plaque rupture. Intriguingly, eroded plaques are more frequently seen in younger patients and females, compared to plaque rupture which is commonly seen in older patients and those with CVD risk factors [51]. These differences in patient epidemiology may yield insight into the underlying mechanisms which determine the fate of the plaques.

## 4. Are Glucocorticoids Athero-Protective or Drivers of Lesion Development?

The cumulative evidence gathered over the past 20 years highlighting increased atherosclerosis in Cushing’s patients suggests that glucocorticoids may be pathophysiological drivers of plaque development. Individuals with Cushing’s exhibit higher circulating LDL levels, a thickened intimal-medial layer, and a more narrowed lumen in the carotid artery compared to age- and sex-matched non-Cushing’s individuals [52]. A more recent meta-analysis investigating atherosclerosis in Cushing’s also concluded that individuals with Cushing’s Syndrome have a thickened intimal-medial layer and larger plaques in their carotid artery, along with impaired vasodilation compared to control non-Cushing’s individuals [53]. Notably, this vascular remodelling occurs independent of several cardiovascular/atherosclerosis risk factors including smoking, body mass index, blood pressure, glucose levels and lipid levels. Indeed, when compared with healthy and hypertensive individuals, patients with Cushing’s show increased carotid intima-media thickness and a higher prevalence of plaques (26% vs. 0 and 16%, respectively) [54]. Cushing’s patients also demonstrated increased ankle-brachial index (ABI) compared to controls, a clinical marker of peripheral vascular disease (PVD) [54]. Strikingly, this evidence of increased atheroma remains beyond the remission of Cushing’s when compared to non-Cushing’s individuals, indicating irreversible consequences of systemic changes [19,55]. The increased risk of CVD in these patients, during and beyond remission, highlights the importance of understanding the mechanisms underlying the role of glucocorticoids in atherosclerosis.

The lesion burden of lower limb PVD patients is greater in individuals that have had long term (>5 years) glucocorticoid treatment compared to individuals that have plaques but no previous exposure to exogenous glucocorticoid [56]. In particular, SLE and RA patients have accelerated atherosclerosis which is thought to be due to their glucocorticoid treatment [57,58]. However, it is important to recognise that individuals receiving glucocorticoid therapy often have significant underlying inflammatory conditions. Indeed, independent of glucocorticoid therapy, the increased cardiovascular risk of these SLE and RA patients is thought to be attributed to underlying systemic inflammation, endothelial cell dysfunction and vascular remodelling [59,60,61]. While these studies have demonstrated a positive correlation between glucocorticoids and plaque development, a causal relationship remains an area of active study in pre-clinical models.

Although chronic, supraphysiological exposure to glucocorticoids increases the risk and burden of atherosclerosis in humans, the earliest evidence of glucocorticoid action in pre-clinical models indicated a protective role. In cholesterol-fed rabbits, co-administration of “low dose” dexamethasone (0.125 mg/day) protected against further progression of atherosclerosis, with reduced plaque development, macrophage recruitment and subsequent foam cell formation [62,63]. In mice with atherosclerosis, both acute (5 weeks) and chronic (17 weeks) high dose corticosterone administration (50 mg/mL in drinking water) during development of atherosclerosis decreased lesion size compared with vehicle controls [64]. Notably, chronic administration of corticosterone was accompanied by a marked decrease of macrophage content in lesions (56% vs. 27% in acute treatment). This suggests long-term glucocorticoid exposure over the course of atheroma development can be athero-protective by reducing monocyte/macrophage recruitment and foam cell formation. Indeed, this is supported by a recent study demonstrating the direct ability of prednisone/prednisolone to reduce cholesterol accumulation in macrophages [65].

In contrast, a number of pre-clinical studies employing manipulations in endogenous glucocorticoid levels, often subtle in comparison to the pharmacological administration seen above, have demonstrated pro-atherogenic effects of glucocorticoids. In ApoE^−/−^ mice, a chronic increase in endogenous corticosterone levels in response to repeated stressful stimuli increased the prevalence of lesions compared to unstressed mice [66]. Similarly, genetic manipulation of endogenous corticosterone in mouse models of atherosclerosis supports a pro-atherogenic role. Mice lacking the glucocorticoid inactivating enzyme 11β-HSD2 display accelerated atherosclerosis development when crossed with ApoE^−/−^ mice, compared to controls [67], while mice lacking the enzyme responsible for generating active corticosterone from inactive 11-dehydro corticosterone in tissues, namely 11β-HSD1, have reduced plaque size and severity on an ApoE^−/−^ background compared to controls [68]. Intriguingly, and in contrast to the effects of exogenous glucocorticoid administration, the cellular mechanism underlying the reduction in atherosclerosis in these models has been attributed to altered cholesterol flux in macrophages within lesions, resulting in increased foam cell formation [68,69].

From an overview standpoint there appears compelling evidence for glucocorticoids to act in both an adverse and beneficial manner in the context of atherosclerosis. However, the question of whether glucocorticoids can be viewed as drivers of atherosclerosis or anti-atherogenic is more likely too simplistic an approach. Rather, it is becoming clear that the key to understanding this apparent paradox requires a multifactorial approach, and lies in identifying the crucial cellular targets of glucocorticoids in the vasculature compared to the broad targets resulting from systemic administration, and in parallel understanding the mechanisms that such cells utilise to regulate their response to glucocorticoids.

## 5. Glucocorticoid-Mediated Regulation of Endothelial Cell Dysfunction

The importance of glucocorticoids in maintaining vascular reactivity is well established, and they are vital for regulation of blood pressure and blood flow [70,71]. However, these crucial physiological effects can become adverse in the long-term, as seen in the pathogenesis of Cushing’s-induced hypertension [72]. This evidence implies an important role for glucocorticoids in contributing to endothelial cell dysfunction.

Glucocorticoids can influence vascular reactivity through modulating either vasoconstriction or vasodilation (Figure 2). Endothelin-1 (ET-1) and angiotensin-II (AngII) are two well-studied vasoconstrictors regulated by glucocorticoids. ET-1 acts on ET_A_ and ET_B_ receptors in a paracrine manner on vSMCs to cause contraction, and in an autocrine manner on endothelial cells to cause relaxation [73]. In human atherosclerosis, circulating ET-1 levels are increased, in parallel with increased expression of ET-1 receptor subtypes in atherosclerotic lesions [74,75]. Early in vitro studies in both rat and rabbit vSMCs demonstrated the ability of glucocorticoids to increase ET-1 mediated contractility through increased ET-1 production [76,77], an effect shown to be GR-dependent [78]. The production of AngII is controlled through the action of angiotensin converting enzyme (ACE), converting AngI to the active vasoconstrictor AngII. In response to low blood pressure, endothelial cell production of ACE enhances local generation of AngII, causing vasoconstriction and restoring normal blood pressure [79]. However, excessive ACE-mediated AngII production can result in inappropriate vasoconstriction, contributing to endothelial cell dysfunction. Fishel et al. demonstrated that exposure of cultured rat vSMCs to dexamethasone increased both ACE mRNA and activity [80]. These findings were subsequently extended to human vSMCs and expanded upon to show that induction of ACE led to increased activation of AngII-mediated intracellular signalling via the phospholipase C pathway, stimulating intracellular Ca^2+^-mediated contraction [81]. Moreover, glucocorticoids can further enhance vasoconstriction by inducing AngII type I (AT I) receptors in vSMCs, enhancing AngII signalling [82].

Glucocorticoids can also regulate vascular tone through modulation of vasodilation. In particular, they can inhibit the synthesis and release of the vasodilators nitric oxide (NO) and prostacyclin (PGI_2_) from endothelial cells [83]. The maintenance of NO is key for physiological function, with imbalances in NO contributing to both atherosclerosis and hypertension. Dexamethasone has been shown to promote reactive oxygen species (ROS) overproduction in human endothelial cells through action on the mitochondrial electron transport chain, NAD(P)H oxidase and xanthine oxidase [84], reducing the bioavailability of NO and causing oxidative stress, contributing to endothelial cell damage in the vasculature. Moreover, acute in vivo exposure to dexamethasone in rats blocks the synthesis of NO in the endothelium by inhibiting endothelial nitric oxide synthase (eNOS) [85], which is proposed to be a result of transcriptional repression and increased mRNA degradation [70]. Taken together, glucocorticoids can directly induce endothelial cell dysfunction through hypercontractility and inhibition of vasodilation which may contribute to the initiation of atherosclerosis. On the other hand, there is contrasting evidence demonstrating that high dose glucocorticoids may promote the synthesis of vasodilators. Exposure to dexamethasone can activate eNOS in human endothelial cells through non-genomic mechanisms, reducing vascular inflammation in a mouse model of vascular injury [86]. In a murine model of myocardial infarction, dexamethasone increased basal eNOS activity by 39% [86]. While species- and model-specific differences may play a role in these contrasting results. The higher concentration of dexamethasone used in the murine study compared to the rat study [85], suggests dose-dependent effects, with acute low dose exposure to glucocorticoids inhibiting vasodilator production, whereas high dose glucocorticoid exposure may have vasodilatory effects.

## 6. Effects of Glucocorticoids on Vascular Remodelling

As with endothelial cell dysfunction, glucocorticoids have been shown to have contrasting athero-protective and pro-atherogenic effects on cells involved in vascular remodelling during lesion formation (Figure 3). Glucocorticoid actions on endothelial cells are widely accepted, downregulating their expression of adhesion molecules including ICAM-1, VCAM-1 and E-selectin, and thus reducing recruitment of monocytes and neutrophils [87,88,89]. Moreover, glucocorticoids inhibit endothelial cell production of pro-inflammatory cytokines including IL-6, CXCL8 (IL-8) and CCL2 (MCP-1) [90,91]. Together, these actions are likely beneficial in atherosclerosis by dampening the immune response.

However, evidence indicates that glucocorticoid-mediated responses in both vSMCs and macrophages may depend on the experimental model, as well as the level of glucocorticoid exposure. Glucocorticoids can have potentially atheroprotective effects on vSMCs by inhibiting their migration and proliferation, key events underlying intimal thickening and vascular remodelling. Initial studies in a rabbit carotid collar vascular injury model found that dexamethasone administration 24 h prior to collar-induced vascular injury inhibited intimal thickening, a result of glucocorticoid-mediated inhibition of vSMC migration and proliferation [92]. In rats, dexamethasone has been demonstrated to suppress aortic SMC proliferation by inhibiting phosphorylation of Rb, a checkpoint for the G1 phase of mitosis [93,94], and through inhibition of PDGF-induced migration, mediated by upregulation of PGC-1α expression [95]. In human vSMCs isolated from atherosclerotic arteries, several synthetic glucocorticoids (dexamethasone, hydrocortisone, prednisolone) show clear inhibitory effects on vSMC migration in vitro [96]. In contrast, the ability of glucocorticoids to induce ET-1 production in vSMCs is known to promote vSMC migration [97,98].

During atherogenesis, vSMC migration is dependent upon MMP-mediated breakdown of the ECM to facilitate movement. Specifically, MMP-2 and MMP-9 are increased after vascular injury. Dexamethasone treatment inhibits MMP-2 activity and migration in rat vSMCs; however, this effect was not observed in human aortic vSMCs [99]. Additionally, cortisol inhibited MMP-2 secretion in a human macrophage cell line [100]. These differing outcomes may indicate species-specific responses of vSMC to glucocorticoids but may also be a result of the experimental paradigm; in their respective studies dexamethasone and cortisol were administered for 30 min vs. 24 h. The individual contribution of each of these actions to atherosclerotic plaque development is difficult to dissect. However, it is important to consider that, in the context of vSMC migration and proliferation, potential beneficial effects resulting from reduced migration must also be considered in light of plaque stability, since vSMC are major components of the “protective” fibrotic cap.

Aside from an inhibitory role on monocyte recruitment, there is evidence of multiple, often contrasting, effects of glucocorticoid on macrophages during atherosclerotic plaque development. In rabbits, systemic dexamethasone treatment reduced macrophage accumulation in the intimal and medial layers of the femoral artery, suggesting a protective role of glucocorticoids in vascular remodelling [101]. Indeed, this would support established literature demonstrating the ability of glucocorticoids to inhibit the release of pro-inflammatory mediators such as TNF-α and IL-12 from macrophages [102], thus dampening their pro-inflammatory response. However, targeted delivery of prednisolone to macrophages via liposomes throughout disease progression in LDLR^−/−^ mice accelerated atherosclerosis, with increased macrophage content and larger necrotic cores [103]. This paradoxical pro-atherogenic effect was believed to result from glucocorticoid-induced macrophage lipotoxicity. In line with a lack of a protective effect of glucocorticoids on atherosclerosis when targeted directly to macrophages, a similar study in humans delivering prednisolone in liposomes found that while these liposomes accumulated in macrophages of atherosclerotic femoral lesions, they showed none of the expected anti-inflammatory effects [104]. Instead, further evidence supports a role for glucocorticoids in driving atherogenesis by promoting macrophage lipotoxicity. Specifically, glucocorticoids have been shown to stimulate cholesterol esterification and suppress HDL efflux both in human vSMCs and in macrophages in a GR-dependent manner [105,106]. However, these studies explored glucocorticoid influence on intracellular cholesterol trafficking. A recent study has demonstrated the ability of glucocorticoids to reduce oxLDL uptake in human macrophages by suppressing expression of genes involved in influx, potentially limiting progression of atherosclerosis by preventing accumulation of lipids [107].

In summary, glucocorticoids can play a role in endothelial cell dysfunction and vascular remodelling, with the potential to induce effects that either drive or oppose the development of atherosclerosis (illustrated in Table 2). To further understand these conflicting effects of glucocorticoids at the vascular level it is important to understand control of local availability of glucocorticoid in specific tissues.

## 7. Mechanisms Controlling Glucocorticoid Bioavailability at the Vasculature

Intriguingly, the contrasting literature on glucocorticoid-mediated effects on macrophages highlights both a crucial consideration in assessing conclusions drawn from past research, and an important avenue of future research; namely, the effect of systemic glucocorticoids vs. local glucocorticoid action. Glucocorticoid activity at the vascular wall is dependent not only on the expression of its cognate receptor, but also on the bioavailability of the steroid itself and its access to intracellular receptors. Consequently, numerous cell-specific mechanisms must be considered when assessing the influence of glucocorticoids in atherosclerosis.

### 7.1. Pre-Receptor Metabolism of Glucocorticoids

The importance of local regulation of glucocorticoids at the vascular wall by the 11β-hydroxysteroid dehydrogenase (11β-HSD) enzymes has been extensively reviewed [108,109,110]. Indeed, research over the past 20 years has demonstrated the importance of such pre-receptor glucocorticoid metabolism by the 11β-HSD isozymes in mediating tissue-specific availability of glucocorticoid. The 11β-HSD type 2 (11β-HSD2) isozyme catalyses the inactivation of cortisol to cortisone (corticosterone to 11-dehydrocorticosterone in rodents), whereas 11β-HSD type 1 (11β-HSD1) catalyses the reverse reaction, activating cortisone to cortisol (11-dehydrocorticosterone to corticosterone in rodents) [111]. Both enzymes are present in the vasculature, with 11β-HSD1 mainly expressed in vSMCs in humans and rodents, and 11β-HSD2 is expressed mainly in endothelial cells [112,113]. This pre-receptor metabolism controls downstream transcription, and so cell-specific expression of these metabolising enzymes plays a key role in determining glucocorticoid action in these cells. Indeed, it is hypothesised that 11β-HSD1 is heavily involved in vascular remodelling and angiogenesis, whereas 11β-HSD2 protects against inappropriate glucocorticoid-mediated inhibition of vasodilation [114]. Evidence supporting a pathological role for 11β-HSD1 in atherosclerosis is found in the increased expression of 11β-HSD1 in lipid-storing vSMCs within plaques compared to “non-pathological” contractile vSMCs [115]. Moreover, 11β-HSD1 is increased in aorta samples obtained during surgery from metabolic syndrome patients that have active coronary atherosclerosis [116]. These changes indicate that increased local glucocorticoid action within the vasculature may be important in vascular remodelling and the development of atherosclerosis.

Pre-clinical studies using genetic deletion and pharmacological inhibition of 11β-HSD1 implicate this enzyme as a driver of atherosclerosis [67]. The inhibition of 11β-HSD1 in ApoE^−/−^ mice reduced neointimal thickening and increased collagen content, promoting plaque stability [117]. In a similar model, 11β-HSD1 inhibition prevented plaque progression and reduced aortic lesion size in parallel with an improved lipid profile [118]. Two independently generated 11β-HSD1 knockout strains crossed with ApoE^−/−^ mice demonstrated similar protection from atherosclerosis [68,69]. In the study by Kipari et al., 11β-HSD1 deficiency resulted in macrophages with increased cholesterol ester export, implicating a role for glucocorticoids in increasing the cholesterol ester content of macrophages and thus contributing to foam cell formation [69]. Intriguingly, this study also demonstrated that 11β-HSD1 deficiency specifically in marrow-derived cells confers atheroprotection, indicating the potential importance of 11β-HSD1 in macrophages and/or related leukocytes. However, despite preclinical studies showing that 11β-HSD1 inhibition is atheroprotective, no clinical studies of 11β-HSD1 inhibitors have been conducted for the treatment of atherosclerosis [109]. This may be due to the lack efficacy of 11β-HSD1 inhibitors for the treatment of obesity and type 2 diabetes mellitus which has resulted in them failing to progress beyond phase II clinical trials [119]. Thus, there remains an unmet clinical need for identifying alternative mechanisms through which the body may control glucocorticoid action in these cells.

### 7.2. Cognate Receptor Distribution and Regulation

The majority of glucocorticoid effects are a consequence of transcriptional actions mediated through intracellular binding to either the low-affinity GR or the high-affinity MR [7,8,9]. Upon ligand binding in the cytosolic space, the receptor–ligand complex dissociates from chaperone proteins and translocates to the nucleus, where it binds directly or indirectly to the promoter region of target genes to either activate or repress transcription. In contrast to the ubiquitous expression of GR, MR expression is limited to relatively few tissues, with the kidney being the classical target for MR activation. Extra-renal MR has a higher affinity for cortisol than aldosterone [8], and thus preventing aberrant MR activation by glucocorticoids, which circulate at concentrations 100–1000 times higher than aldosterone, is dependent upon the co-expression of 11β-HSD2. Therefore, the response of cells to glucocorticoids depends on the expression of both the receptors and pre-metabolising enzymes. For example, in the hippocampus and in adipose tissue, expression of both GR and MR in the absence of 11β-HSD2 often results in inverted U-shaped relationships between glucocorticoid concentrations and transcriptional outcomes as specific receptors become activated and saturated [108,120]. In the vasculature, while GR and MR are expressed both in endothelial cells and in vSMCs, GR levels are significantly greater in both cell types, with the same being true in macrophages [67,108]. The expression of 11β-HSD2 both in endothelial cells and in vSMCs suggests inactivation of glucocorticoids prevents signalling via MR [67]. Indeed, the importance of this “roadblock” to MR activation was elegantly demonstrated when ApoE^−/−^ mice were crossed with 11β-HSD2^−/−^ mice. These mice exhibited increased development of severe atheromatous plaques even in that absence of high-fat/Western diet, a mechanism attributed to glucocorticoid-mediated activation of MR [67]. In a separate AAV-PCSK9 murine model of atherosclerosis, deletion of MR from endothelial cells resulted in reduced plaque inflammation, attributed to downregulation of endothelial cell adhesion molecule expression and thus reduced leukocyte infiltration [121]. Interestingly, this reduction in inflammation was only seen in male mice, not females, suggesting a complex sex difference of the MR on atherosclerosis development in mice.

In contrast to endothelial MR activation appearing to promote atherosclerosis, evidence indicates an athero-protective role for GR activation. Endothelial cell-specific GR deletion results in more severe atherosclerosis and increased recruitment of macrophages in an ApoE^−/−^ model of atherosclerosis [122]. However, this athero-protective effect appears to be cell-specific, as activation of GR in macrophages has been shown to contribute to atherogenesis. Macrophage-specific GR deletion in an LDLR^−/−^ mouse model of atherosclerosis did not alter lesion size or macrophage content [123]. Interestingly these mice did display a reduction in vascular calcification in the lesion, implying a role for GR-activation in macrophages in contributing to calcification in atherosclerosis. What remains to be determined is a role for GR and/or MR dysregulation in mediating beneficial or adverse glucocorticoid action with respect to atherosclerotic disease progression.

### 7.3. Control of Glucocorticoid Delivery to the Vasculature: A Role for CBG?

Under physiological conditions, CBG binds 80–90% of circulating glucocorticoids, with 2–10% remaining unbound. Consequently, in line with the free hormone hypothesis, this means that only between 2 and 10% of glucocorticoids are able to diffuse into tissues and cells and exert biological activity. Recent genome-wide association studies have demonstrated the importance of CBG in regulating circulating levels, with genetic variation in the *SERPINA6/1* locus influencing hepatic *SERPINA6* expression associated with morning plasma cortisol [35,124].

Unsurprisingly, given the role of glucocorticoids in numerous physiological settings, CBG has been postulated to play a significant role in glucocorticoid action in several pathological conditions. For example, in sepsis, a reduction in plasma CBG levels gives rise to an increased free fraction of circulating cortisol in the circulation [125]; an important anti-inflammatory response to enable the body to fight the systemic infection. Pre-clinical studies with genetically modified mice have also indicated roles for CBG in the delivery of glucocorticoids to tissue. CBG deficient mice are more vulnerable to LPS-induced sepsis, and exhibit reduced glucocorticoid action in tissues such as liver and brain [126]. To our knowledge, changes in CBG expression in atherosclerosis have yet to be evaluated in either pre-clinical models or in humans. However, several factors known to regulate CBG levels are altered in atherosclerosis. In particular, IL-6, which is elevated in early inflammation, potentiates the inflammatory response and is implicated in lipid metabolism and plaque formation [127], has been shown to inhibit CBG mRNA and protein expression in vitro. Several studies have shown that increased plasma IL-6 is associated with a decrease in CBG levels [128], with in vitro studies demonstrating that IL-6 inhibits CBG mRNA and protein levels [129]. This implies that an IL-6-mediated reduction in CBG may allow more “free” cortisol to act at the vasculature in atherosclerosis in an attempt to dampen the inflammatory response. Yet, as discussed previously, prolonged rises in free cortisol may become maladaptive and potentiate plaque development.

However, it is not only alterations in the levels of CBG which can influence glucocorticoid bioavailability. Structural changes in CBG protein are known to regulate its affinity for glucocorticoids. In particular, proteolytic cleavage of the reactive certain loop of CBG is known to result in an approximately 10-fold decrease in its binding affinity for cortisol [130,131]. A small number of endogenous proteases are capable of this cleavage, including NE and some MMPs, a mechanism that is thought to enable local release of glucocorticoid at sites of inflammation [4]. NE action on CBG remains the most well studied, and intriguingly, while NE action in atherosclerosis continues to be highlighted, a link between NE-mediated cleavage of CBG and glucocorticoid delivery to the vasculature during atherosclerosis remains to be investigated.

## 8. Conclusions

While it is clear that chronic, systemic glucocorticoid excess is positively associated with increased atherosclerosis, evidence is now being uncovered for a causative role for glucocorticoids in the cardiovascular complications arising from atherosclerotic plaque formation. This has re-ignited the debate around the influence of glucocorticoids in disease progression and treatment. The ability of glucocorticoids to affect multiple cell types within the vasculature, in parallel with mediating opposing actions depending on the concentration and cellular target, makes determination of relative importance difficult. The situation is made more complex by the multiple regulatory mechanisms still being discovered that control glucocorticoid bioavailability, both at sites of plaque development and in specific cells. Pre-clinical studies in genetically modified mice continue to aid our understanding of these mechanisms, identifying potential new therapeutic targets to control glucocorticoid action. However, it is clear that while these mechanisms can alter glucocorticoid action, such changes need to be considered not only in the context of specific cells, but in the context of plaque site, disease progression and treatment. For example, limiting vSMC migration may be viewed as beneficial in limiting necrotic core formation, but it may lead to a fibrous cap more vulnerable to rupture. Given the widespread use of animal models, a greater understanding of the species-specific differences in plaque composition and vulnerability would also be of significant benefit. Together with a better knowledge of the mechanisms controlling local, endogenous glucocorticoid action, these insights will be key in both predicting the response of patients to systemic glucocorticoid treatment, and in advancing personalised, targeted glucocorticoid therapy to combat cardiovascular disease.

## Figures and Tables

**Figure 1 ijms-22-07622-f001:**
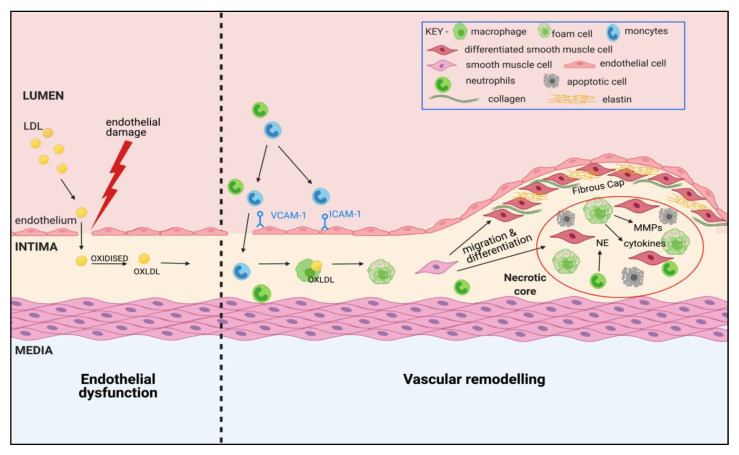
Pathogenesis of atherosclerotic lesions with features of vulnerability. In response to damage to the endothelium, circulating LDL-cholesterol infiltrates the intimal layer and becomes oxidised by free radicals into oxidised LDL (oxLDL), triggering an inflammatory cascade in which immune cells (including monocytes and neutrophils) are recruited from the circulation by attachment to adhesion molecules (VCAM-1 and ICAM-1) on the surface of the endothelium. Infiltrated monocytes differentiate into pro-inflammatory macrophages and ingest oxLDL, forming foam cells. Smooth muscle cells migrate from the medial layer and differentiate to form a fibrous cap containing collagen and elastin. Proteases including matrix metalloproteinases (MMPs) and neutrophil elastase (NE) are released from immune cells, resulting in degradation of the extracellular matrix, facilitating vascular remodelling and enhancing the pro-inflammatory mileu. The accumulation of foam cells, apoptotic/necrotic cells and migrated smooth muscle cells forms a necrotic core, which upon rupture is highly thrombotic.

**Figure 2 ijms-22-07622-f002:**
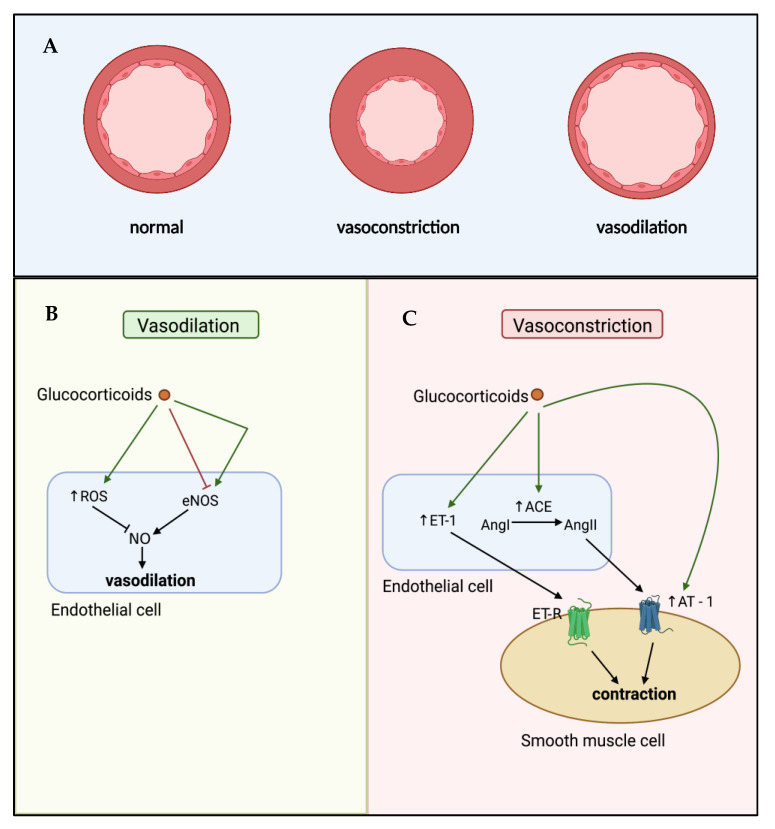
Glucocorticoid-mediated effects on vascular contractility. (**A**) Cross-section of an artery demonstrating comparative differences during normal conditions, under vasoconstriction, and under vasodilation. (**B**) Glucocorticoids can both inhibit and promote vasodilation in endothelial cells. Glucocorticoids can increase reactive oxygen species (ROS), which act to reduce nitric oxide (NO) availability, a key driver of vasodilation. Glucocorticoids can also inhibit NO production by reducing endothelial nitric oxide synthase (eNOS). However, there is conflicting data indicating that, under certain conditions (e.g., high dose), glucocorticoids can induce eNOS, driving NO-mediated vasodilation. (**C**) Glucocorticoids can promote vascular contractility in smooth muscle cells through increasing production and activity of the vasoconstrictors endothelin-1 (ET-1) and Angiotensin II (AngII), the latter through increased expression of angiotensin-converting enzyme (ACE) and the Angiotensin II type 1 receptor (AT-1). ET-R—Endothelin-1 receptors; AngI—angiotensin I.

**Figure 3 ijms-22-07622-f003:**
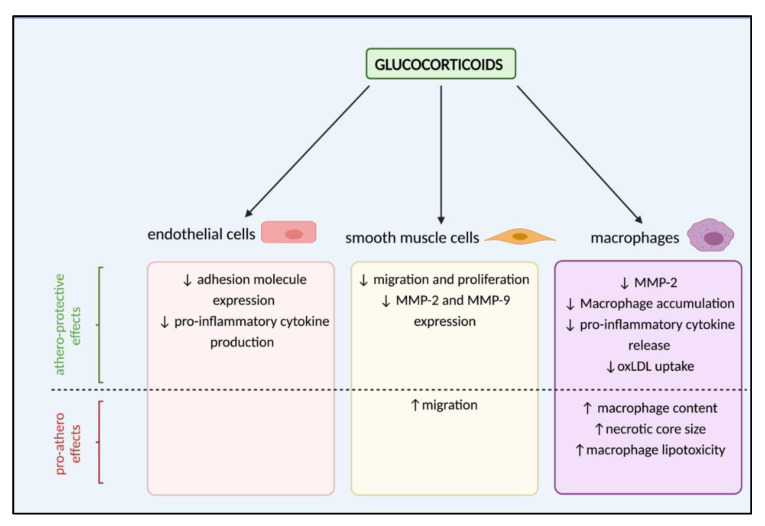
Summary of cell-specific effects of glucocorticoids during atherosclerotic plaque development. Evidence indicates that glucocorticoids are capable of exerting both athero-protective and pro-atherogenic (pro-athero) effects in several crucial cell types. In endothelial cells, glucocorticoid action is athero-protective. However, in both smooth muscle cells and macrophages, evidence exists of both pro-athero and athero-protective effects. oxLDL—oxidised LDL; MMP—matrix metalloproteinases.

**Table 1 ijms-22-07622-t001:** Glucocorticoid action in cardiovascular health: when adaptive becomes maladaptive.

Acute, Adaptive Physiological Roles	Chronic, Maladaptive Effects	Pathological Outcomes	References
Regulation of glucose homeostasis	Hyperglycaemia	Diabetes mellitus	[11]
	Insulin resistance		
Maintenance of energy homeostasis	Visceral adiposity	Obesity	[12]
	General weight gain		
Maintenance of vascular tone and blood pressure	Impaired vasodilation	Hypertension	[13]
	Increased contractility		
	Plasma volume expansion		
Regulation of lipid metabolism	Elevated cholesterol	Dyslipidaemia	[14]
	Elevated triglycerides		
Heart development and function	Bradycardia Cardiac hypertrophy	Heart failure	[13]
Regulation of clotting factors and fibrinogen	Hypercoagulability	Thrombosis	[15,16]

Glucocorticoids act in an acute manner on multiple physiological systems to restore homeostasis. Chronic exposure to excess glucocorticoid turns adaptive responses into maladaptive outcomes, resulting in pathologies linked to the development of cardiovascular disease.

**Table 2 ijms-22-07622-t002:** Actions of glucocorticoids in the vessel wall that may inhibit or enhance atherosclerotic lesion formation.

Potential Impact	Target	Effect	Reference
Pro-Atherosclerotic	Endothelium	↓ generation of EDRFs	[83]
	Endothelium	↑ ROS generation	[84]
	Endothelium	↓ eNOS	[85]
	vSMC	↑ vasoconstrictor (ET-1) generation	[78]
	vSMC	↑ vasoconstrictor (AngII) generation	[80,81]
	vSMC	↑ AT-1 receptor expression	[82]
	vSMC	↑ accumulation of cholesterol esters	[106]
	Macrophages	↑ lipotoxicity	[104]
	Macrophages	↑ accumulation of cholesterol esters	[105]
Anti-Atherosclerotic	Endothelium	↑ eNOS (non-genomic)	[86]
	Endothelium	↓ adhesion molecule expression	[87,88,89]
	Endothelium	↓ expression of pro-inflammatory cytokines	[90,91]
	Macrophages	↓ accumulation	[101]
	Macrophages	↓ expression of pro-inflammatory cytokines	[102]
	Macrophages	↓ oxLDL uptake and cholesterol efflux	[107]
Unclear	vSMCs	↓ proliferation and migration	[92,93,94,95,96,99]

Glucocorticoids can exert effects on cells involved in the formation of atherosclerotic lesions that may be predicted to either promote or inhibit lesion formation. The impacts of some glucocorticoid-mediated actions are more difficult to predict as they may be regarded as both beneficial and detrimental to plaque development. For example, inhibition of smooth muscle cell migration may reduce lesion size but increase vulnerability to rupture. Glucocorticoids can have opposing effects dependent on species and type/duration of glucocorticoid exposure. AngII—angiotensin II; EDRF—endothelium-derived relaxing factor; eNOS—endothelial nitric oxide synthase; ET-1—Endothelin-1; ROS—reactive oxygen species; vSMC—vascular smooth muscle cell.

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
