# Peer review of "Glucocorticoids: Fuelling the Fire of Atherosclerosis or Therapeutic Extinguishers?"

_ijms, 2021, doi:10.3390/ijms22147622_

Round 1
Reviewer 1 Report
This manuscript by MacLeod et al. presented a comprehensive summary of the roles glucocorticoids play in the pathology of atherosclerosis. The manuscript is well organized, very informative, and fluent in language/style. Therefore the reviewer would suggest the acceptance of the manuscript with some very minor changes: 1) there are a couple of typos in the manuscript, e.g. Line 391, "importantavenue" - there should be a space between the two words. Please read through the manuscript carefully during the revision; 2) it would be nice to have a table summarizing the roles glucocorticoids play in atherosclerosis, separating the beneficial and detrimental sides of each role, along with references. This would tremendously help the readers to identify the information needed.
Author Response
Detailed list of revisions and responses to reviewer comments
The authors would like to express our thanks to both reviewers for their positive and supportive comments, and also for making insightful suggestions for ways to improve the manuscript. We have followed the suggested corrections and additions highlighted by the reviewers and feel that these have improved the quality of our manuscript.
Response to Reviewer 1:
This manuscript by MacLeod et al. presented a comprehensive summary of the roles glucocorticoids play in the pathology of atherosclerosis. The manuscript is well organized, very informative and fluent in language/ style.
Thank you for these positive comments.
Therefore the reviewer would suggest the acceptance of the manuscript with some very minor changes:
1) there are a couple of typos in the manuscript, e.g. Line 391, "importantavenue" - there should be a space between the two words. Please read through the manuscript carefully during the revision;
Thank you for this suggestion. We have re-read the manuscript in an attempt to remove any remaining typographical and grammatical errors. Please see below for a full list of corrections.
2) it would be nice to have a table summarizing the roles glucocorticoids play in atherosclerosis, separating the beneficial and detrimental sides of each role, along with references. This would tremendously help the readers to identify the information needed.
The authors thought this was a very helpful suggestion and have inserted a new table (Table 2) to fulfil this role. To introduce this table we have also added in an extra section of text:
Lines 476-480
In summary, glucocorticoids can play a role in endothelial cell dysfunction and vascular remodelling, with the potential to induce effects that either drive or oppose the development of atherosclerosis (illustrated in table 2). To further understand these conflicting effects of glucocorticoids at the vascular level it is important to understand control of local availability of glucocorticoid in specific tissues.

Reviewer 2 Report
Dear editors,
MacLeod and colleagues present a review on the topic of glucocorticoids in the development of atherosclerosis. The review is beautifully written, interesting and makes it clear to the reader that this topic is complex and that there is a clear gap in our understanding when it comes to GC metabolism and cardiovascular disease.
I only have a minor recommendation which the authors may have a look at, that I believe will make the manuscript even better and more appealing.
For the introduction, I’d like to refer the authors to PMID 23312287 in which the authors show that cortisol follows an excretion pattern that goes beyond circadian rhythmicity (in response to changing levels of salt). If the authors were so inclined, this could serve to show that measuring cortisol adequately is even more complicated than we thought.
I think it would be great to add subclinical hypercortisolism to the manuscript [e.g. Review PMID: 25871954: On the one hand, the authors present Cushing’s as a very rare disease that clearly shows what elevated cortisol levels can do to a person. Then you have iatrogenic Cushing’s syndrome, which is a necessary evil so to speak. Now as a clinician I would think „well, Cushing’s rare and people with SLE or other auto-inflammatory diseases need to be treated, and if we have to resort to high-dose glucocorticoids it’s still better than the alternative.“ Now it would be really great to show that there might be hypercortisolism that is somewhat lurking in the shadows. If subclinical hypercortisolism is a thing and cortisol is pro-atherogenic in humans (one way or the other), then this is significant. And it suddenly changes from exotic or inevitable to something that might be more common than one would think.
Major issues
- none
Minor issues
- see above
Typos
- Mechanisms controlling glucocorticoid bioavailability at the vasculature
- 1. pre-receptor metabolism
Line 432: „studies […] have no conducted […]“
- 2. cognate receptor distribution
- 3. control of glucocorticoid delivery
Line 487: „in sepsis were […]“
Author Response
Detailed list of revisions and responses to reviewer comments
The authors would like to express our thanks to both reviewers for their positive and supportive comments, and also for making insightful suggestions for ways to improve the manuscript. We have followed the suggested corrections and additions highlighted by the reviewers and feel that these have improved the quality of our manuscript.
Response to Reviewer 2:
MacLeod and colleagues present a review on the topic of glucocorticoids in the development of atherosclerosis. The review is beautifully written, interesting and makes it clear to the reader that this topic is complex and that there is a clear gap in our understanding when it comes to GC metabolism and cardiovascular disease.
Thank you for these very positive comments; they are much appreciated.
I only have a minor recommendation which the authors may have a look at, that I believe will make the manuscript even better and more appealing.
For the introduction, I’d like to refer the authors to PMID 23312287 in which the authors show that cortisol follows an excretion pattern that goes beyond circadian rhythmicity (in response to changing levels of salt). If the authors were so inclined, this could serve to show that measuring cortisol adequately I even more complicated than we thought.
This was a great suggestion; we have addressed it by adding the following text:
Page 2, lines 35-37.
Additionally, there is also evidence of longer infradian rhythms of cortisol excretion suggesting more complex control than just circadian rhythmicity, which may add further complication to accurate measurement of exposure to cortisol [2].
I think it would be great to add sub-clinical hypercortisolism to the manuscript [e.g. Review PMID: 25871954: On the one hand, the authors present Cushing’s as a very rare disease that clearly shows what elevated cortisol levels can do to a person. Then you have iatrogenic Cushing’s syndrome, which is a necessary evil so to speak. Now as a clinician I would think „well, Cushing’s rare and people with SLE or other auto-inflammatory diseases need to be treated, and if we have to resort to high-dose glucocorticoids it’s still better than the alternative.“ Now it would be really great to show that there might be hypercortisolism that is somewhat lurking in the shadows. If subclinical hypercortisolism is a thing and cortisol is pro-atherogenic in humans (one way or the other), then this is significant. And it suddenly changes from exotic or inevitable to something that might be more common than one would think.
Again, the authors thought this was a great suggestion and have added additional text to address this point (see below). In addition, we took the opportunity to augment this be referring to the potential role of ‘tissue-specific’ Cushing’s syndrome in the development of atherosclerosis.
Pages 3-4 (lines 85-117).
Whilst cases of Cushing’s are extremely rare, affecting between 40 to 70 people out of every million [20], the link between glucocorticoid excess and cardiovascular risk has been identified in other scenarios. Sub-clinical hypercortisolism (also referred to as ‘subclinical Cushing’s syndrome’, ‘pre-clinical Cushing’s syndrome’, or ‘sub-clinical autonomous glucocorticoid hypersecretion’) is an imperfectly defined condition indicated in patients exhibiting mild alteration in cortisol secretion without the symptoms of Cushing’s syndrome [21,22]. It is usually associated with adrenocortical incidetalomas detected during imaging for conditions unrelated to the adrenal gland. Clinical investigations have associated chronic exposure to ‘sub-clinical’ hypercortisolism with a number of complications (including excess weght/obesity, hypertension, and diabetes mellitus) that are risk factors for CVD [22,23]. Indeed, there is some evidence of increased cardiovascular events and mortality in these patients, indicating a possible benefit from medical intervention or adrenalectomy [24]. This is important as adrenal incidentalomas are considerably more common than Cushing’s syndrome (possibly present in 4-7% of adults) [25], and the prevalence of sub-clinical hypercortisolism in these patients has been estimated as 0.2-2.0% [23].
A further link between glucocorticoid excess and the development of cardiovascular disease has been been suggested in patients with metabolic syndrome; a condition associated with increased cardiovascular risk [26]. The similarity between the features of metabolic syndrome (a combination of at least three symptoms out of: central obesity, hypertension, hyperglycaemia, hypertriglyceridaemia, and low serum high-density lipoprotein) and Cushing’s syndrome, has prompted the suggestion that the former represents a ‘low cortisol’, or ‘tissue-specific’ form of Cushing’s syndrome [27]. This proposal is based on an improved understanding of the role of glucocorticoid-metabolising enzymes (specifically the 11β-hydroxysteroid dehydrogenases (11β-HSDs)) in regulating activation of corticosteroid receptors in a tissue-specific manner: raising the possibility of tissue-specific glucocorticoid excess through aberrant expression of the 11β-HSD isozymes (e.g. in the liver [28] or central adipose tissue [29]) without elevated circulating levels of the steroid. This has led to the investigation pharmacological targetting of 11β-HSDs for treatment of metabolic disease.
In addition to the evidence obtained from exposure to elevated endogenous glucocorticoids, the widespread therapeutic use of glucocorticoids to treat inflammatory and autoimmune conditions has also provided supporting evidence for their association with CVD. Revisions made (tracked changes) –
Grammatical/sentence structure changes
Line 5 “…Queen’s Medical…” changed to “The Queen’s Medical…”
Line 19 “…dependent upon the multifactorial actions of glucocorticoids changed to “…dependent upon their multifactorial actions .”
Line 44 – ‘glucocorticoid. [2].’ changed to ‘glucocorticoid [3].’
Line 171 – ‘Response-to-injury hypothesis’ changed to ‘response-to-injury hypothesis’
Line 175 – ‘Monocytes in particular’ change to ‘In particular, monocytes’
Line 195 – ‘the aggregation inflammatory cells’ changed to ‘the aggregation of inflammatory cells’
Line 209 – ‘NE expression is increased’ changed to ‘The expression of NE is increased’
Line 227 ‘…a distinct mechanism different from plaque rupture [50].’ changed to ‘…a distinct mechanism from plaque rupture [50].’
Line 243 – ‘higher circulating LDL’ changed to ‘higher circulating LDL levels’
Line 258 ‘…importance to understand…’ changed to ‘…importance of understanding…’
Line 305 – ‘in both and adverse and a beneficial manner’ changed to ‘in both an adverse and a beneficial manner’
Line 313 ‘5. Glucocorticoid regulation of Endothelial Dysfunction’ changed to ‘5. Glucocorticoid-mediated regulation of Endothelial Cell Dysfunction’
Line 317 – ‘This evidence implied’ changed to ‘This evidence implies’
Line 348 ‘…comparitive differences during normal conditions, under vasocontrsiction…’ changed to ‘…comparative differences during normal conditions, under vasoconstriction…’
Line 352 ‘…there is conflciting data indicating that under certain conditions (e.g. high dose), glucocortiocids can…’ changed to ‘…there is conflicting data indicating that, under certain conditions (e.g. high dose), glucocorticoids can…’
Line 395 ‘Dexamethasone exposure…’ changed to ‘Exposure to dexamethasone…’
Line 398 – ‘results, the’ changed to ‘results. The’
Line 411 ‘ … VCAM-1 and E-selectin on…’ changed to ‘…VCAM-1 and E-selectin,…’
Line 419 ‘…glucocortoicd…’ changed to ‘…glucocorticoid…’
Line 428 ‘…to collar vascular injury inhibited…’ changed to ‘…to collar-induced vascular injury inhibited…’
Lines 447-449 ‘…beneficial effects resulting from reduced migration must also be considered in light of plaque stability, of which vSMC are major components of the ‘protective’ fibrotic cap.’ changed to ‘…potential beneficial effects resulting from reduced migration must also be considered in light of plaque stability, since vSMC are major components of the ‘protective’ fibrotic cap.’
Line 471 – ‘trafficking, a’ changed to ‘trafficking. A’
Line 495 – changed ‘importantavenue’ to ‘important avenue’
Lines 538-539 - studies of 11β-HSD1 inhibitors have not conducted changed to no clinical studies of 11β-HSD1 inhibitors have been conducted (in response to reviewer 2)
Line 549 – ‘of atherosclerosis. did not alter lesion’ changed to ‘of atherosclerosis did not alter lesion’
Line 562 ‘…with the same is true in macrophages…’ changed to ‘…with the same being true in macrophages…’
Line 598 – ‘For example in sepsis were a reduction in plasma CBG’ changed to ‘For example, in sepsis, a reduction in plasma CBG levels gives rise to’ (in response to reviewer 2)
Line 606 - 608 – ‘atherosclerosis, in particular, IL-6, which is elevated in early inflammation, potentiating the inflammatory response and is implicated in lipid metabolism and plaque formation [117] has been shown to inhibit CBG mRNA and protein expression in vitro.’
Changed to
‘In particular, IL-6, which is elevated in early inflammation, potentiates the inflammatory response and is implicated in lipid metabolism and plaque formation [117], has been shown to inhibit CBG mRNA and protein expression in vitro.’
Line 611 – ‘Additionally, in atherosclerosis, IL-6 This implies that an IL-6’ changed to ‘protein levels [119]. This implies that an IL-6’
Line 621- ‘small number of endogenous proteases are capable this cleavage, including’ changed to ‘A small number of endogenous proteases are capable of this cleavage, including’
Line 642 – ‘For example limiting’ changed to ‘For example, limiting’